# Structural Insights into the Dynamics of Water in SOD1 Catalysis and Drug Interactions

**DOI:** 10.3390/ijms26094228

**Published:** 2025-04-29

**Authors:** Ilkin Yapici, Arda Gorkem Tokur, Belgin Sever, Halilibrahim Ciftci, Ayse Nazli Basak, Hasan DeMirci

**Affiliations:** 1Department of Molecular Biology and Genetics, Koc University, Istanbul 34450, Türkiye; iyapici21@ku.edu.tr (I.Y.); atokur20@ku.edu.tr (A.G.T.); 2Department of Pharmaceutical Chemistry, Faculty of Pharmacy, Anadolu University, Eskisehir 26470, Türkiye; belginsever@anadolu.edu.tr; 3Department of Molecular Biology and Genetics, Burdur Mehmet Akif Ersoy University, Burdur 15030, Türkiye; hciftci@mehmetakif.edu.tr; 4Suna and İnan Kıraç Foundation, Neurodegeneration Research Laboratory (KUTTAM-NDAL), School of Medicine, Koc University, Istanbul 34450, Türkiye; 5Stanford PULSE Institute, SLAC National Laboratory, Menlo Park, CA 94025, USA

**Keywords:** ALS, hSOD1, X-ray crystallography, structural biology, drug design, molecular modeling, radical scavenger, Abl1 inhibitors

## Abstract

Superoxide dismutase 1 (SOD1) is a crucial enzyme that protects cells from oxidative damage by converting superoxide radicals into H_2_O_2_ and O_2_. This detoxification process, essential for cellular homeostasis, relies on a precisely orchestrated catalytic mechanism involving the copper cation, while the zinc cation contributes to the structural integrity of the enzyme. This study presents the 2.3 Å crystal structure of human SOD1 (PDB ID: 9IYK), revealing an assembly of six homodimers and twelve distinct active sites. The water molecules form a complex hydrogen-bonding network that drives proton transfer and sustains active site dynamics. Our structure also uncovers subtle conformational changes that highlight the intrinsic flexibility of SOD1, which is essential for its function. Additionally, we observe how these dynamic structural features may be linked to pathological mutations associated with amyotrophic lateral sclerosis (ALS). By advancing our understanding of hSOD1’s mechanistic intricacies and the influence of water coordination, this study offers valuable insights for developing therapeutic strategies targeting ALS. Our structure’s unique conformations and active site interactions illuminate new facets of hSOD1 function, underscoring the critical role of structural dynamics in enzyme catalysis. Moreover, we conducted a molecular docking analysis using SOD1 for potential radical scavengers and Abelson non-receptor tyrosine kinase (c-Abl, Abl1) inhibitors targeting misfolded SOD1 aggregation along with oxidative stress and apoptosis, respectively. The results showed that CHEMBL1075867, a free radical scavenger derivative, showed the most promising docking results and interactions at the binding site of hSOD1, highlighting its promising role for further studies against SOD1-mediated ALS.

## 1. Introduction

Oxidative stress, a major cause of cellular damage, occurs when cells produce superoxide (O_2_^−^) radicals as a by-product of oxygen reduction during cellular respiration [1]. Cu-Zn superoxide dismutase 1 (SOD1), which catalyzes the conversion of two O_2_^−^ anions into H_2_O_2_ and O_2_, operates mainly in the cytoplasm and plays a central role as a cytoprotectant against oxidative stress [2]. By eliminating these highly reactive O_2_^−^ species, SOD1 acts as a frontline defense in preventing oxidative damage to cellular components such as proteins, lipids, and nucleic acids [3].

Human SOD1 (hSOD1) is a highly conserved 32 kDa homodimeric metalloenzyme, with each monomer containing a zinc cation (Zn^2+^) for structural stability and a copper cation (Cu^2+^) for catalytic activity [4]. The imidazole ring of His63 acts as a ligand to both cations, bridging them together [5,6]. Zn^2+^ is additionally coordinated by two histidine residues (His71 and His80) and an aspartic acid residue (Asp83), while five-coordinate Cu^2+^ relies on three more histidine residues (His46, His48, and His120) and a water molecule [7]. Not only the precise coordination of metal ions but also their dynamic interplay with key residues, O_2_^−^ anions, and water molecules within the active site are crucial in the conversion of radicals.

The well-established ping-pong reaction mechanism of hSOD1 involves the sequential reduction and reoxidation of copper by two distinct O_2_^−^ anions [8,9]. Structural studies of SOD1 across various eukaryotic organisms provide additional support for this finely tuned catalytic cycle [6,10,11,12,13]. The O_2_^−^ anion forms H bonds with the Arg143 side chain, replacing the water coordinated to Cu^2+^ [14,15]. The reduction of Cu^2+^ to Cu^1+^ by the lone pair electrons of the O_2_^−^ radical releases O_2_ and breaks the bond between copper and His63, creating a tri-coordinated Cu^1+^ state [16]. The now protonated histidine H-bonds with another O_2_^−^ anion, which also interacts with Arg143. Through bond rearrangements and proton addition, H_2_O_2_ is formed and Cu^1+^ is oxidized back to Cu^2+^, restoring its coordination with His63 [10,17]. The water molecules’ coordination, positioning, and dynamic behavior within the active site channel are critical as they orchestrate the necessary proton transfer for catalysis [18,19]. Despite the identification of water molecules, a more comprehensive understanding of their contribution to proton transfer, metal ion coordination, and overall catalytic efficiency is necessary to fully elucidate hSOD1’s function.

Single-point mutations in hSOD1 are linked to its gain of neurotoxicity in amyotrophic lateral sclerosis (ALS), the third most frequent, yet still orphan, neurodegenerative disease [20,21]. While only ~5–10% of ALS cases are familial, over 200 mutations in hSOD1, encoded by just 153 amino acids, are the culprit behind ~20% of inherited and ~1–2% of sporadic ALS [22,23]. These mutations frequently preserve enzymatic activity but destabilize the protein, causing it to misfold and form toxic aggregates, leading to predominantly upper and lower motor neuron degeneration [24]. Although SOD1-specific Tofersen and the two other FDA-approved drugs for ALS (edaravone and riluzole) can slow disease progression and extend survival, they are unable to reverse its course [25,26,27]. Thus, a deeper understanding of the structure–dynamics–function relationship of this scavenger protein is essential for developing next-generation therapeutic strategies that address the underlying causes of ALS, rather than merely alleviating symptoms.

Phenol derivatives are known radical scavengers with hydroxyl groups directly attached to an aromatic benzene ring [28]. However, they are generally not considered as therapeutics because they have irritating, corrosive, and highly toxic features [29]. To mimic the radical scavenging activity of phenol through its hydroxybenzene moiety, Mitsubishi Tanabe Pharma Corporation developed potential phenol-like radical scavengers bearing a carbonyl group that can be easily converted into a hydroxyl group by keto-enol tautomerization [20,30]. Among a number of compounds, edaravone (Figure 1) was identified as an effective radical scavenger capable of scavenging both lipid- and water-soluble peroxyl radicals [20,30,31]. Edaravone possesses neuroprotective and anti-inflammatory effects against oxidative stress and against activated microglial cells, respectively [32]. Edaravone has been reported to improve motor functions in SOD1^G93A^ mutant mice [33] and SOD1^H46R^ mutant rats [34]. Ito et al., 2008 [33], determined the in vivo efficacy of edaravone for possible future treatment of ALS with the first randomized blind study. Their findings indicated that 15 mg/kg edaravone administration lowered the rate of reduction in parameters such as body weight, mean persistence time on the Rotarod, and grip power in SOD1^G93A^ mutant mice. One of the most striking points they reached was that the mean area of abnormal SOD1 deposition per anterior horn and its root exit zone decreased significantly. In another study, Aoki et al., 2011 [34], reported that edaravone improved motor functions in SOD1^H46R^ mutant male rats with 5.6 ± 0.5 cm mean landing foot-splay distance between the hind feet in a high-dose group. Edaravone received FDA approval for the treatment of ALS in 2017 [20].

Gallic acid, 3,4,5-trihydroxybenzoic acid (Figure 1), is a strong antioxidant and radical scavenger. Gallic acid can protect biological cells, tissues, and organs against oxidative damage caused by reactive oxygen species (ROS) [35]. Excess ROS production due to the oxidation of dopamine may trigger neurodegenerative disorders. The antioxidant and neuroprotective effects of gallic acid and its derivatives have been investigated in particular in the treatment of neurodegenerative disorders such as Alzheimer’s and Parkinson’s diseases [36,37,38,39,40]. Baek et al. have also found the ability of gallic acid to interfere with the formation of SOD1 filaments that could be effective in the treatment of ALS. They also showed that gallic acids were able to bind to the soluble SOD protein (12–44 μM kd), indicating its direct binding to the SOD1 protein [41].

Reactive carbonyl species (RCS) are cytotoxic products of oxidative stress that damage proteins, nucleic acids, and lipids [42]. Among the subgroups of RCS, α,β-unsaturated aldehydes such as 4-hydroxy-trans-2-nonenal (4-HNE) are highly cytotoxic derivatives [43]. Since 4-HNE has been identified as a contributor to delayed neuronal death by apoptosis, compounds with 4-HNE scavenging properties have been investigated for the prevention of delayed neuronal death [44,45]. Noguchi et al., 2019 [46], reported that L-carnosine hydrazide (L-carnosine-NHNH_2_, CNN) (Figure 1) exhibited protective effects in PC-12 cells with a 30 mM concentration towards 250 μM 4-HNE and abolished delayed neuronal death by >60% in the pre-treated group and by >40% in the post-treated group.

Apoptosis is another potential mechanism of motor neuron death in ALS [47,48,49]. Recent studies indicated that the activation of Abelson non-receptor tyrosine kinase (c-Abl, Abl1) was detected in mutant SOD1 transgenic mice models [50]. The high levels of Abl gene expression have been reported to stimulate apoptosis in motor neurons and inflammation in astrocytes and microglia, causing neurodegeneration and neuroinflammation [51,52,53,54]. During oxidative stress, Abl1 overexpression causes neuronal apoptosis as it is involved in the regulation of the maximal activation of p53, which stimulates growth arrest and apoptosis as a tumor suppressor in response to DNA damage [55]. Dasatinib (Figure 1) showed neuroprotection in SOD1^G93A^ transgenic ALS mice, reducing the cytotoxicity of mutant SOD1 significantly based on cell viability and cell death assays (*p* < 0.05). Dasatinib also improved the survival, weight loss, and poor grip strength of SOD1^G93A^ mice at a dose of 25 mg/(kg·day) compared with a vehicle treatment (*p* < 0.01, 25 mg/(kg·day)) [50]. Imatinib (Figure 1) showed the micromolar inhibition of Abl1 phosphorylation in primary neuron cultures in response to oxidative stress. Rojas et al., 2015 [56], revealed that 2 μM imatinib along with astrocyte conditioned media (ACM)-SOD1^G93A^ alleviated the intensity of immunoreactivity for phosphorylated c-Abl and the number of positively stained cells [56]. Imamura et al., 2017 [57], performed high-throughput screening of numerous compounds using the survival of ALS patient iPSC-derived motor neurons as a readout. They determined that more than half of the hits targeted the Src/c-Abl signaling pathway, and they identified that bosutinib (Figure 1) increased ALS motor neuron survival (mean survival: 164.1 ± 9.4 days compared to vehicle (156.3 ± 8.5 days)) and regulated misfolded SOD1 proteins in transgenic mice.

In this study, we determined the high-resolution crystal structure of hSOD1 in a previously unobserved crystal form and packing, revealing an assembly of six homodimers composed of twelve monomers. The unprecedented arrangement of these dimers and their active sites offers a unique opportunity to observe how structural dynamics are modulated by the surrounding solvent environment, particularly with respect to the behavior of coordinated water molecules. These coordinated water molecules are steered into alternative conformations, positions, and coordination states within the twelve catalytic sites, creating an intricate choreography. A deeper look into the dynamics of the coordinated water molecules and the active site enhances our understanding of the catalytic mechanism of the intrinsically dynamic SOD1. By capturing this dynamic water-mediated choreography, our structure provides fresh insights into how small conformational changes in hSOD1 can influence its overall function and stability, directly relevant to its role in ALS. In addition, to determine the affinity and possible binding modes of hSOD1, we have performed molecular modelling studies for radical scavengers (edaravone, gallic acid, and CNN) and Abl1 inhibitors (dasatinib, imatinib, and bosutinib) and their derivatives at the binding site of hSOD1 (PDB ID: 9IYK).

## 2. Results

### 2.1. Crystal Structure of hSOD1 in a New Crystal Form

Using X-ray synchrotron cryocrystallography, we determined the crystal structure of recombinant wild-type hSOD1 (please see the Materials and Methods section, Section 4.1, for the sequence information) at a 2.3 Å resolution in the monoclinic C121 space group, with unit cell parameters *a* = 178.19 Å, *b* = 138.25 Å, *c* = 112.93 Å, *α* = 90°, *β* = 129.2°, and *γ* = 90°. The data collection and refinement statistics are summarized in Table 1. The asymmetric unit of the crystal consists of six hSOD1 homodimers (A-B, C-D, E-F, G-H, I-J, and K-L chains), comprising twelve independently refined monomers (Figure 2A). The electron densities of chains B and D allowed the modeling of 156 residues. The remaining monomers were modeled with 153 residues, except for chains G and K, which lack the density of N-terminal alanine. A total of seventy-five alternative side-chain conformations were built into the overall model. In addition to 1 Zn^2+^ and 1 Cu^2+^ cation per monomer in the metal-binding sites and a total of 503 water molecules, the final model also includes 4 sulfate and 4 acetate anions originating from the crystallization mother liquor (Figure 2C).

The Zn^2+^ cation supplemented prior to the crystallization formed a center of symmetry (Zn^CRYST^) and glued four different chains in a near-tetragonal geometry with angles of 95.03°, 90.99°, 84.18°, and 97.95° (Figure 2C,D). During purification, the hexahistidine tag of the recombinant hSOD1 was cleaved using thrombin protease following nickel affinity chromatography. However, residues Ser-2, His-1, and Met0 in chains B and D remained as artefacts, with their electron densities clearly visible at the 1σ level (Appendix A). The Zn^CRYST^ anchors these residual His-1 residues in chains B and D, as well as the otherwise flexible His110 residues in chains A and C (Figure 2D, Appendix A). The stabilization of metal-coordinated His110 residues by Zn^CRYST^ results in lower individual B-factors and better-defined electron densities compared to the free His110 residues in other chains (Appendix A).

Ellipsoid representations of the B-factors reveal that the acidic core of the crystal structure is more rigid. In contrast, the solvent-exposed peripheral regions display increased flexibility, illustrating a gradient of decreasing rigidity from the core to the periphery (Figure 2B, Appendix A). Chain H has the highest average B-factor of all atoms, reaching 96.1 Å^2^, and the weakest electron density, making it the most flexible monomer out of twelve in the asymmetric unit. Chains A and B, with B-factors of 53.0 and 55.1 Å^2^, respectively, form the most rigid dimeric pair among those adopting the canonical dimer interface (Appendix A).

### 2.2. hSOD1 Monomers and Dimers Display Modest Conformational Changes

To analyze the structural variations among the asymmetric chains, each homodimer and monomer was superposed (Figure 3A and Figure 4A). The superposition of dimer pairs resulted in Root Mean Square Deviation (RMSD) values ranging from 0.32 Å to 0.68 Å (Figure 3B). Similarly, monomer superposition resulted in RMSD values between 0.28 Å and 0.72 Å, revealing subtle conformational differences among individual chains based on Cα atom alignments (Figure 4B). These observations suggest a modest degree of structural plasticity and heterogeneity.

Pairwise alignments were performed to explore the positional differences in equivalent Cα atoms and plotted against the residue numbers (Figure 3C and Figure 4C). The A-B dimeric pair was used as the reference for dimers, while chain A served as the reference for monomers. In the dimeric comparison, monomers on the left side are referred to as dimer_A_, while those on the right side are denoted as dimer_B_.

Notable structural differences were observed in loop II (residues 23–27) and the Greek key loop, loop VI (residues 102–115), particularly around residues Asn26 and His110. Interestingly, the striking fluctuation of His110, reaching up to 2.4 Å in monomers and 2.3 Å in dimer_A_, is not observed in dimer_B_, where the RMSD score is less than 1 Å. The positional shift of His110 is most likely caused by its coordination with Zn^CRYST^ of reference chain A. As expected, Asn26, His110, and their neighboring residues exhibit consistently high B-factors, distinguishing them from the rest of the structure. The averaged B-factors across the monomers are 100.9 ± 26.8 Å^2^ for Gly108, 120.4 ± 24.5 Å^2^ for Asp109, 104.6 ± 34.9 Å^2^ for His110, and 117.7 ± 18.4 Å^2^ for Asn26, all of which significantly exceed the overall average of 69.6 Å^2^. This trend is uniformly observed across all chains, highlighting the intrinsic flexibility of these residues (Appendix A).

The three artefact residues left after thrombin cleavage of the N-terminal hexahistidine tag in chains B and D, positioned near the natural N-terminal Ala1, affect its position. Among them, His-1 plays a primary role in reinforcing this positional shift through its coordination with Zn^CRYST^, resulting in a deviation of up to 7 Å^2^ compared to other chains. The electrostatic loop (residues 122–144), especially around Gly130, Asn131, and Glu132, exhibits a moderate RMSD shift, reaching up to 1.4 Å in dimer_B_. The elevated individual B-factors within loop I (10–13), reaching up to 157.3 Å^2^ at residue 11, indicate a distinctive disorder in chain J, resulting in a positional difference of up to 1.6 Å, a variation not observed in any other chain. Although the impact of crystal lattice contacts cannot be excluded, these slight fluctuations may be attributed to the intrinsic plasticity of the hSOD1 enzyme.

### 2.3. Dynamic Water Molecules Reveal New Conformations of the Active Site

Water molecules are primarily steered into the Cu^2+^ active site through highly complex mechanistic steps of H bonding and coordination chemistry (Figure 5). Initially, the copper active site predominantly adopts a distorted tetrahedral geometry, where His46 coordinates through its ND1 atom, while His48, His63, and His120 coordinate through their NE2 atoms (Figure 5A,B). W1 forms H bonds with other water molecules, first with W2 (Figure 5C) and then with W3, W4, and W5 (Figure 5D). When W3’s electron cloud is sufficiently close to the His120 residue (~3.4 Å, Appendix A), W3 coordinates with this histidine residue (Figure 5D). After W3-His120 coordination, W1 coordinates with His63 residue (Figure 5E). These coordinations can be further validated by the bond length of Cu^2+^ and π-nitrogen atoms of His120 residues being shortened (1.8 Å) since coordination between τ-nitrogen atoms and the W2 molecule increases the electron density of the π-nitrogen atom that can strengthen the coordinated covalent bond (Appendix A). His120 is proposed to be one of the water-steering residues guiding the H-bonded water molecules into the active site since the coordination length between W1 and the His63 residue diminishes while the His63-Cu site length decreases and the His120-Cu site length elongates (Figure 5E). After W1 approaches the copper active site, W3 loses its coordination with the His120 residue. In this mechanistic step, we observed that water molecules can remain in different configurations with various coordinations and bond lengths (Figure 5F–I). When W1 is sufficiently close to the His63 residue (2.4 Å, Appendix A), W1 coordinates with Cu^2+^ to form the water-bound, catalytically native-like form of the SOD1, which causes elongation and significant weakening on the Cu-His63 bond as is apparent by the evident break in the dissociated electron densities (Figure 5J). After the water-bound form is achieved, strong coordination between the π-nitrogen atom of His63 and Cu^2+^ is restored as electron densities re-overlap with each other (Figure 5K). Finally, the thermodynamically most stable, meshwork water structure forms (Figure 5L).

### 2.4. Arg143 Dynamics Reveal a Key Allosteric State

Our crystallographic data suggest that water molecules mediate allosteric alterations through interactions with the Arg143 residue (Figure 6, Appendix A). The guanidino group of Arg143 forms H bonds with the hydroxyl residue of Thr58 and consistently with the backbone of Cys57 and Gly61, thereby maintaining the integrity of the disulfide loop throughout the structure (Figure 6A). As W1 approaches the guanidinium group of Arg143, it coordinates with the ε- and η1-nitrogen atoms of this group (Figure 6B). After coordinating with several other water molecules, all ε- and η-nitrogen atoms are surrounded by these coordinated water networks (Figure 6C,D). This coordinated water network alters the whole H bond network between the Arg143, Cys57, Thr58, and Gly61, either through bond elongations and shortenings or bond formations and breakings. As an example, the H bond between the hydroxyl residue of Thr58 and η2-nitrogen of Arg143 is broken, whilst the H bond between His48 and η1-nitrogen of Arg143 is repositioned by the formation of H bonds among His48 and η2-nitrogen of Arg143 (Figure 6D). This repositioning effect on the structure can be further demonstrated by the simultaneous rearrangement in the Cu active site, which causes W3-His120 H bond formation, indicating that this repositioning is the allostericity-driver step (Figure 5D). This may play a crucial role in its mechanism as the coordination of the water network with Arg143 likely decreases the electron density on the ε- and η-nitrogen atoms that cause the weakening of H bonds with Cys57, Thr58, and Gly61 residues and cause bond elongations and breakages. After this step, the water network starts to dissolve, and only the η-nitrogen atoms of Arg143 are coordinated with water molecules (Figure 6E,F). When W1 coordinates with W3 and W4 and approaches the η1-nitrogen atom, the H bond between His48 and Arg143 is restored (Figure 6G). When this network is rearranged such that coordination within W4, ε- and η1-nitrogen atoms are formed, the electron density of the η1-nitrogen atom and ε-nitrogen atom decreases, causing an accumulation of electron density on the η2-nitrogen atom and restoration of the H bond between Thr58 and the Arg143 guanidino group (Figure 6H,I). Subsequently, allostericity-driver water molecules start to move further from the arginine residue, and coordination between W1 and Arg143 η1-nitrogen atoms becomes broken (Figure 6J,K) and recruits new water molecules to form the highly coordinated final state (Figure 6L). During this time course, when W1-Arg143 coordination is broken, Cu and W1 coordination is formed in the active site (Figure 5J). Due to the highly dense H bond network around Arg143, the H bond between His48 and Arg143 becomes broken again (Figure 6L). Moreover, as W1, W2, and W3 move further from the allosteric Arg143 site, a coordination network on the active site is formed by these water molecules (Figure 5L).

### 2.5. Docking Studies of 9IYK Structure

Based on the potential of radical scavengers (edaravone, gallic acid, and CNN) and Abl1 inhibitors (dasatinib, imatinib, and bosutinib) for mitigating the levels of unwanted SOD1 formation, we performed a molecular docking assessment for edaravone, gallic acid, CNN, dasatinib, imatinib, and bosutinib and their derivatives in the binding pocket of hSOD1 (PDB ID: 9IYK). We retrieved 128 edaravone, 110 gallic acid, 108 CNN, 57 dasatinib, 57 bosutinib, and 55 imatinib derivatives using ChEMBL database (https://www.ebi.ac.uk/chembl/, accessed on 14 January 2025). The results showed that many derivatives of edaravone, gallic acid, CNN, dasatinib, bosutinib, and imatinib generally show higher affinity than standard agents. The docking scores of hit edaravone derivatives (CHEMBL4205125, CHEMBL1490540, CHEMBL1390061, CHEMBL1518895, and CHEMBL1872892) were found to be −7.641, −7.002, −6.865, −6.767, −6.484 kcal/mol, respectively (Table 2), compared to edaravone (−4.831 kcal/mol) (Table 2). These derivatives interacted with Asn65, Glu49, His63, His80, Ser68, and Lys136 (Figure 7A). CHEMBL4205125 formed H bonds with Asn65, and π cations and salt bridges with Lys136 (Figure 7B).

Radical scavengers gallic acid (−5.428 kcal/mol) (Table 2) and CNN (−5.769 kcal/mol) (Table 2) displayed higher affinity than edaravone in the binding pocket of hSOD1. Among the gallic acid derivatives, CHEMBL1256379, CHEMBL339835, CHEMBL2146905, CHEMBL47027, and CHEMBL95308 exhibited the most significant interactions with Asn65, His63, His80, Glu49, Lys136, Ser68, and Thr135 through H bonding, π cation formation, and salt bridge formation (Figure 8A,B). Gallic acid also interacted with Zn202 via salt bridge formation. The docking scores were calculated as −7.121, −7.027, −6.794, −6.778, and −6.761 kcal/mol (Table 2) for CHEMBL1256379, CHEMBL339835, CHEMBL2146905, CHEMBL47027, and CHEMBL95308, respectively.

On the other hand, CNN derivatives showed the highest docking scores compared to the edaravone and gallic acid derivatives. CHEMBL1075867, CHEMBL429344, CHEMBL1075871, CHEMBL1075979, and CHEMBL1076109 demonstrated the highest affinities, with the docking scores of −9.760, −8.935, −8.858, −8.841, and −8.757 kcal/mol, respectively (Table 2) and interactions with Asn65, His63, His80, Glu49, Glu132, Lys136, Ser68, and Thr135 (Figure 9A,B).

The molecular docking results showed that Abl1 inhibitors (dasatinib, bosutinib, and imatinib) generally presented lower affinities compared to radical scavengers (edaravone, gallic acid, and CNN), with docking scores of −5.751, −4.382, and −4.969 kcal/mol, respectively (Table 2). Among the derivatives of Abl1 inhibitors, the most promising affinity was observed with dasatinib derivatives. The dasatinib derivatives CHEMBL3425716, CHEMBL4451230, CHEMBL4439020, CHEMBL4865214, and CHEMBL5283559 were identified as hit derivatives, with docking scores of −7.244, −7.231, −7.164, −6.972, −6.889 kcal/mol, respectively (Table 2). These derivatives established interactions with Asn65, Arg143, Asp96, Glu100, Glu132, Glu133, Lys70, Lys136, and Trp32 (Figure 10A,B).

The docking results of hit bosutinib derivatives, CHEMBL198211, CHEMBL339893, CHEMBL124333, CHEMBL128326, and CHEMBL197930, showed scores of −6.459, −6.291, −6.166, −6.156, and −6.142 kcal/mol, respectively (Table 2). They formed crucial H bonds, π cations, π stacks, and salt bridges with Asn65, Glu133, His80, Lys70, and Lys136 (Figure 11A,B).

The hit imatinib derivatives were identified as CHEMBL4794277, CHEMBL4740754, CHEMBL3098615, CHEMBL4763826, and CHEMBL1765716, with docking scores of −6.527, −6.491, −6.489, −6.488, and −6.193 kcal/mol, respectively (Table 2). Asn65, Asp96, Glu100, Glu132, Glu133, His80, Lys136, and Thr135 were the key residues with which these derivatives interacted (Figure 12A,B).

## 3. Discussion

Our 2.3 Å crystal structure of wild-type hSOD1 reveals a previously unreported crystal form, further expanding the diversity of known SOD1 crystallographic arrangements [59]. Fully metalated;, disulfide-containing; and dimeric, eukaryotic SOD1 is exceptionally stable, retaining dismutase activity even in 4 M guanidine-HCl or 8 M urea [6,60,61,62], yet it is also remarkably dynamic. Even in our 2.3 Å cryo-crystallographic structure at 100 K, hSOD1 visibly demonstrates its flexibility, displaying structural variation across the six dimers and twelve monomers within the same asymmetric unit. This structural framework provides an exceptional opportunity to gain deeper insights into the catalytic site and the dynamic behavior of the surrounding water molecules.

The crystal structure presented in our work has a non-canonical zinc coordination site, where Zn^CRYST^ glues residual His-1 residues from chains B and D and His110 residues of chains A and C. This zinc-mediated dimer–dimer interaction arises in crystallo, predominantly driven by exogenous Zn^2+^ and the artefactual His-1 residue originating from the recombinant N-terminal hexahistidine tag. As a result, removing either His-1 or excess Zn^2+^ would likely disrupt this specific crystallographic arrangement. Nevertheless, this observation emphasizes the intrinsic capacity of SOD1 to engage in non-native metal coordination, highlighting structural flexibility and potential transient interactions, relevant under physiological or pathological conditions. Importantly, residues participating in this zinc coordination, particularly the surface-exposed His110, are distant from the active site. Thus, the coordinated water dynamics within the catalytic region remain undisturbed, preserving the integrity of the observed water-mediated conformational changes.

His110 has been observed to participate in the coordination of external metal-binding sites in the crystallographic structures of mutant SOD1 but has not been documented to do so in any other wild-type structure. In the monomeric hSOD1 mutant F50E/G51E/E133Q, crystallized in the orthorhombic P212121 space group (PDB ID: 1MFM), His110 directly coordinates one of the nine cadmium cations through its imidazole ring [63]. In the crystal structure of the double copper-binding site mutant H46R/H48Q, crystallized in the P21 space group (PDB ID: 2NNX), two Zn^+2^ cations link the dimers through interactions with a water molecule, Glu77, and His110 [64]. Similarly, in the G85R mutant structures, which affect the metal-binding region and are also crystallized in P21 (PDB IDs: 2VR7, 2VR8), Zn^+2^ cation is coordinated in a tetrahedral geometry by His110 from one dimer, Glu24 from another dimer, and two thiocyanate anions [65]. Our wild-type hSOD1 crystal structure, crystallized in a completely different C121 space group, features Zn^CRYST^ bridging two dimers by gluing the His110 residues of chains A and C, along with the residual His-1 from chains B and D.

Previous studies have demonstrated that metal-catalyzed oxidation (MCO) modifies local metal-binding sites and preferentially targets histidine residues in SOD1 [66]. Since His110 is one of the two non-active site modifications, it has been proposed as a transient metal-binding site [67]. Another study suggests that His110, along with nearby residues Asp109 and Cys111, may contribute to a weak, non-active copper-binding site [68]. Computational analyses further indicate that His110 is one of eight residues that may influence SOD1’s folding kinetics, potentially leading to its aggregation [69]. Although two of the four histidines linked to Zn^CRYST^ are non-native, our structure reveals that His110 in wild-type SOD1, not just in pathogenic mutants, can also interact with a metal cation. Our finding strengthens the possibility that His110 may serve as a transient or weak metal-binding site.

The superposition of different subunits within our structure reveals slight positional differences in the Cα atoms, particularly at Asn26, His110, Gly130, Asn131, and Glu132, residues that are conserved among eukaryotic SOD1s and located in flexible loop regions [13]. Among these, Asn26 and Asn131 are also susceptible to deamidation, a post-translational modification that removes the side-chain amide from asparagine or glutamine residues, subtly altering the local protein environment in a way similar to a missense mutation [70,71]. The flexible electrostatic loop (residues 122–144), which contributes to substrate recognition, plays a crucial role in SOD1 function by mediating key interactions within the active site [72]. Within this region, Glu130 forms H bonds at the N-terminal region of SOD1’s fibrillar structure, while Glu132 is thought to facilitate electrostatic guidance of the O_2_^−^ anion toward the catalytic site [6,73,74]. The flexibility of these residues suggests they may contribute to SOD1’s structural dynamics, enhancing its adaptability and functional regulation, which may explain why none, except for Glu132, have been directly linked to ALS-related pathogenic mutations.

Our study illustrates the interplay between water molecules and SOD1’s active site, where conformational dynamics are modulated by water-mediated interactions and H bonding. The stepwise coordination of water molecules within the copper active site orchestrates a dynamic sequence of conformational changes in the catalytic region, guided primarily by the positioning of the bridging His63 and the copper-coordinating His120. Water molecules play a crucial role in reshaping the active site geometry, transitioning it from a distorted tetrahedral to a trigonal bipyramidal configuration. The fluctuation in Cu^2^⁺-His63 and Cu^2^⁺-His120 bond lengths during this process reflects a continuous exchange between coordination states as water molecules approach and interact with the metal cation.

The coordination dynamics between W1, W3, and the Cu^2^⁺ ion illustrate how subtle changes in water positioning alter the electronic properties of the active site. The elongation of the Cu-His63 bond (3.0 Å) and the shortening of the Cu-His120 bond (1.8 Å) upon W1 interaction reflect how water coordination rearranges the catalytic environment (Appendix A). Beyond enabling the formation of the active enzyme state, water molecules further stabilize these transitions through dynamic H bonding interactions.

Arg143 in the electrostatic loop selectively interacts with anions via electrostatic forces, positioning it as one of the key regulators of local H bond dynamics [7,75]. The orchestrated dynamics of water molecules influence both allosteric regulation and catalytic activity by interacting with Arg143, leading to conformational changes in the active site. The coordinated water network surrounding its guanidino group triggers a cascade of H bond rearrangements, primarily affecting local and metal-binding regions. These disruptions and reformations reshape the H bond network, ultimately modulating the conformation of the metallocenter.

As shown in Figure 5 and Figure 6, the coordinated water network gradually dissolves after inducing the allosteric changes, followed by the recruitment of new water molecules, which leads to the final configuration of the active site. This cyclical process of water coordination and dissociation suggests a dynamic equilibrium where water molecules continuously modulate the SOD1 enzyme’s conformational states, allowing for flexibility and fine-tuning of its catalytic activity. The restoration of the H bond between Arg143 and Thr58, coupled with the reformation of the Cu^2+^-W1 coordination, represents a crucial step in returning the enzyme to its native-like active conformation.

ALS, the third most common neurodegenerative disease after Alzheimer’s and Parkinson’s disorders, leads to degeneration of upper and lower motor neurons in the brain, brainstem, and spinal cord, followed by muscle atrophy. ALS, like other diseases of the same spectrum, currently does not have an effective means of therapy. Although edaravone, riluzole, and tofersen are the only drugs approved by the FDA, their beneficial effects on disease progression are limited [25,26,27,76]. There is an urgent need to discover new therapeutics beyond drug repurposing methods to robustly combat ALS.

Edaravone, a promising radical scavenger for ALS treatment, encouraged us to investigate the anti-ALS effects of other radical scavengers such as gallic acid and CNN due to their promising effects in neurodegeneration. Among edaravone, gallic acid, CNN, and their derivatives, we obtained the best molecular docking results with the CNN derivative CHEMBL1075867. This result indicates that long alkyl chain and benzyl groups contribute significantly to the higher affinity of CHEMBL1075867 compared to that of CNN at the binding site of hSOD1.

Abl1 kinase was detected in mutant SOD1 transgenic mice models provoking neuronal damage and apoptosis in ALS motor neurons. Moreover, high levels in the phosphorylation of Abl1 were observed in mutant SOD1 mice. Dasatinib, bosutinib, and imatinib, which are potential Abl1 inhibitors, were reported to be effective against misfolded SOD1 protein. Therefore, we performed molecular docking studies for dasatinib, bosutinib, imatinib, and their derivatives in the binding pocket of hSOD1. Of these, the most promising results were obtained with dasatinib, while the results of bosutinib and imatinib were found to be similar. The dasatinib derivative CHEMBL3425716 revealed the most significant affinity at the binding site of hSOD1 among the tested derivatives of Abl inhibitors. This outcome indicated that long-chain alkyl esters of CHEMBL3425716 increased the hSOD1 binding potential compared to that of dasatinib.

## 4. Materials and Methods

### 4.1. Expression and Purification of hSOD1

The human SOD1 (hSOD1) construct was designed with the following amino acid sequence to obtain the N-terminally hexahistidine-tagged recombinant protein:

MGSSHHHHHHSSGLVPR(cut)GSHMATKAVCVLKGDGPVQGINFEQKESNGPVKVWGSIKGLTEGLHGFHVHEFGDNTAGCTSAGPHFNPLSRKHGGPKDEERHVGDLGNVTADKDGVADVSIEDSVISLSGDHCIIGRTLVVHEKADDLGKGGNEESTKTGNAGSRLAC GVIGIAQ∗ (∗ is stop). The corresponding gene was codon-optimized and synthesized by Genscript, USA, and cloned into pET28a(+) bacterial expression plasmid by using NdeI and BamHI restriction cleavage sites at its 5′ and 3′ ends, respectively. 

The plasmids were transformed into the competent *Escherichia coli* Rosetta2™ BL21 strain [77]. The transformed cells were grown in LB supplemented with 50 µg/mL kanamycin and 35 µg/mL chloramphenicol at 30 °C overnight. OD_600_ was determined as 0.8 for the induction of protein expression at a final concentration of 0.4 μM IPTG, and the cells were cultured overnight at 20 °C. The cells were centrifuged using the Beckman Allegra 15R desktop centrifuge at 3500 rpm and 4 °C for 30 min for harvesting.

The bacterial cells were dissolved in a lysis buffer containing 50 mM TRIS-HCl (pH 7.5), 500 mM NaCl, and 10 mM imidazole and supplemented with 2 mM 2-mercaptoethanol, 0.1 mM phenylmethylsulfonyl fluoride (PMSF), and 1 mg/mL lysozyme. After, the cells were lysed by sonication with a Branson W250 sonicator (Branson Ultrasonic Corporation, Brookfield, CT, USA). The cell lysate was ultracentrifuged using a Beckman Optima™ L-80 XP (Beckman, Brea, CA, USA) at 35,000 rpm for 1 h at 4 °C by using a Ti-45 rotor (Beckman, Brea, CA, USA). The pellet containing the insoluble debris was discarded, and the supernatant was loaded into the Ni-NTA agarose column after equilibration, at a 2.5 mL/min flow rate (Qiagen, Venlo, The Netherlands). Equilibration and washing were performed with the buffer, containing 20 mM TRIS-HCl (pH 7.5), 200 mM NaCl, and 20 mM imidazole, while the soluble hSOD1 proteins were eluted in a buffer containing 20 mM TRIS-HCl (pH 7.5), 200 mM NaCl, and 250 mM imidazole. The eluted protein was placed in the dialysis membrane of a 3 kDa cut-off and dialyzed against a washing buffer containing 20 mM TRIS-HCl (pH 7.5), 200 mM NaCl, and 20 mM imidazole overnight at 4 °C. In addition, during dialysis, the buffer was supplemented with 1:100 thrombin protease to cleave the N-terminal hexahistidine tag. To collect the hSOD1 without the tag, the protein solution was applied to a Ni-NTA column, and flowthrough without affinity hexahistidine tag was collected. Then, to get rid of impurities, the collected fractions of the untagged hSOD1 were loaded into a Superdex200 column for size-exclusion chromatography with a buffer containing 20 mM TRIS-HCI pH 7.5, and 150 mM NaCl. The pure hSOD1 supplied with 100 µM CuCl_2_·2H_2_O and 100 µM ZnCl_2_ was concentrated by ultrafiltration columns from Millipore (Merck, Darmstadt, Germany) and stored at −80 °C until the crystallization trials.

### 4.2. Crystallization of hSOD1

We employed a sitting-drop micro-batch under the oil screening method for initial crystallization screening, using 72-well Terasaki^TM^ crystallization plates (Greiner-Bio, Frickenhausen, Germany), as explained by Atalay et al., 2023 [78]. Purified hSOD1 protein at 10 mg/mL was mixed at a 1:1 volumetric ratio under ~3500 commercially available sparse matrix crystallization screening conditions. The sitting drop solutions were then covered with 16.6 μL of 100% paraffin oil (Tekkim Kimya, Nïlüfer/Bursa, Türkiye). All the crystallization experiments were performed at ambient temperature. The crystals were obtained at Crystal Screen™ crystallization screen 1 condition #20 (Hampton Research, Aliso Viejo, CA, USA), containing 0.1 M sodium acetate trihydrate (pH 4.6), 0.2 M ammonium sulfate, and 25% *w*/*v* polyethylene glycol 4000. The protein solution and crystallization condition were further screened at a 4:1 volumetric ratio under 48 different conditions from the CryoPro™ screen (Hampton Research, Aliso Viejo, CA, USA), and condition #2 with 6.0 M 1,6-hexanediol yielded the crystals, which were then flash-cooled with liquid nitrogen for cryogenic data collection.

### 4.3. Data Collection, Processing, and Structure Determination of hSOD1

Diffraction data for the hSOD1 were collected at Diamond Light Source (Didcot, UK), beamline I03 (mx-37045) on a Eiger2 XE 16M detector (Dectris, Baden-Daettwil, Switzerland) at 100 K using monochromatic radiation at a wavelength of 0.98 Å, oscillation width of 0.10°, beam size of 80 μm × 20 μm, transmission of 10.00%, and exposure time of 0.0072 s. In total, 3600 images were collected. The data were automatically processed with Xia2 dials [79,80]. The structure was solved via molecular replacement using PHASER v2.8.3 [81] implemented in PHENIX suite [82]. The 1.06 Å resolution structure of the hSOD1 I113T mutant with its ligand (PDB ID: 4A7S) was used as a search model and used for the initial rigid body refinement within the PHENIX software package v1.20 [83]. After simulated-annealing refinement, TLS parameters and individual coordinates were refined. We also performed composite omit map refinement implemented in PHENIX to identify the potential positions of altered side chains and water molecules, which were checked in the COOT v0.9.8.93 [83], and positions with strong difference densities were retained. The data collection and structure refinement statistics are summarized in Table 1.

### 4.4. Molecular Docking

To explore potential radical scavengers (edaravone, gallic acid, and CNN) and Abl1 inhibitors (dasatinib, imatinib, and bosutinib) and their derivatives at the binding site of hSOD1, the ChEMBL database (https://www.ebi.ac.uk/chembl/, accessed on 14 January 2025) was used to obtain the appropriate structures (PDB ID: 9IYK).

The crystal structure of the ambient temperature was acquired from the RCSB database (https://www.rcsb.org/structure/9IYK, accessed on 14 August 2024). The protein preparation module was used to prepare the crude protein for molecular docking. Prime automatically added missing chains, and PropKa calculated the protonation state at physiological pH. SiteMap was then used to identify the highest ranked potential receptor binding sites. The docking grid was determined using grid generation, choosing the hit binding site that compromises the specified residues (Asn26, Asn65, Asn131, Arg143, Asp83, Asp96, Cys57, Glu49, Glu100, Glu132, Glu133, Gly61, Gly130, His46, His48, His63, His71, His80, His120, Lys70, Lys136, Ser68, Thr58, Thr135, Trp32, Cu201, and Zn202). Further docking experiments were performed on the generated grid. Meanwhile, compounds were sketched, cleaned, and prepared with energy minimization using the OPLS 2005 force field at physiological pH via the LigPrep module. The best minimized structures were then subjected to docking experiments without further modification. The flexible ligand alignment tool was used for the superimposition of our structurally similar ligands. After subjecting the resulting ligand to the Glide/SP docking protocols, the same docking procedures were performed for all tested compounds [84,85] (Schrödinger Release 2016-2: Schrödinger, LLC: New York, NY, USA).

## 5. Conclusions

In this study, we determined the high-resolution crystal structure of human SOD1 (hSOD1) at 2.3 Å, capturing an unprecedented view of its active site dynamics, water coordination, and allosteric regulation, while also identifying a non-native external zinc-binding site that bridges multiple chains through His residues. Our findings offer new insights into how water molecules orchestrate key structural transitions, reinforcing their essential role in the enzyme’s catalytic mechanism. By visualizing multiple SOD1 active sites within a single asymmetric unit, we provide a comprehensive picture of water-mediated interactions, particularly in relation to Cu^2^⁺ and Zn^2^⁺ coordination.

One of the most significant findings of this study is the dynamic choreography of water molecules within the active site. We observed that specific water molecules engage in a stepwise exchange, facilitating proton transfer and maintaining the integrity of the metal coordination network. The flexible nature of these interactions underscores the intrinsic adaptability of hSOD1, allowing it to efficiently regulate the conversion of O_2_^−^ radicals under physiological conditions.

Beyond the active site, our data support the hypothesis that Arg143 is a key allosteric regulator, influencing the electrostatic loop and surrounding H bond network. The structural snapshots captured in our study reveal how water molecules dynamically interact with Arg143, modulating its conformation and likely contributing to enzyme stability and function.

Building on our structural analysis, we also explored potential therapeutic avenues by performing molecular docking studies. Among the compounds tested, CHEMBL1075867, a radical scavenger derivative, exhibited the strongest binding to SOD1, outperforming edaravone, the only FDA-approved radical scavenger for ALS. This suggests that structurally optimized scavengers could offer improved neuroprotective effects. Additionally, our docking results highlighted dasatinib derivatives, particularly CHEMBL3425716, as promising candidates for targeting SOD1 misfolding and aggregation. These findings open new directions for drug development efforts aimed at modifying SOD1 behavior in ALS.

Overall, our work provides a more detailed view of hSOD1’s structural landscape, shedding further light on how water coordination, metal ion dynamics, and allosteric regulation contribute to its function. By integrating crystallographic data with computational modeling, we bridge the gap between basic structural biology and therapeutic discovery, offering a foundation for future studies aimed at targeting SOD1 in ALS and other neurodegenerative diseases. By expanding our understanding of SOD1’s molecular mechanisms, we hope to contribute to the development of more effective treatment strategies for ALS and related disorders.

## Figures and Tables

**Figure 1 ijms-26-04228-f001:**
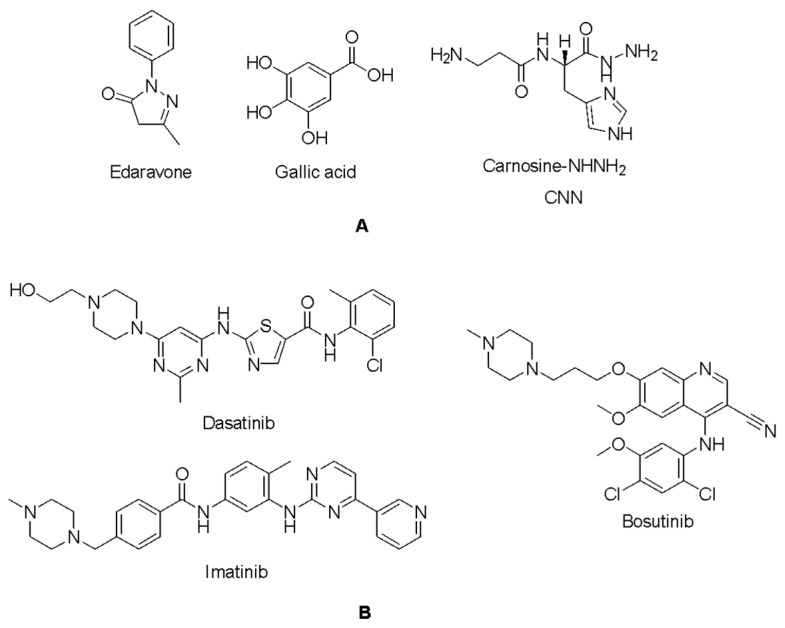
Potential radical scavengers (**A**) and Abl1 inhibitors (**B**) in ALS.

**Figure 2 ijms-26-04228-f002:**
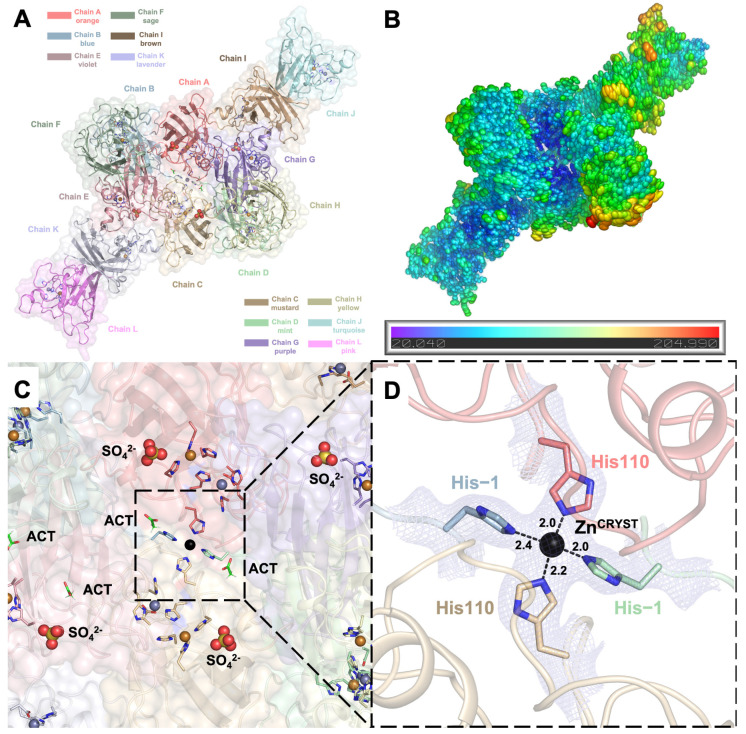
(**A**) The cartoon and semi-transparent surface representations of the hSOD1 crystal structure (PDB ID: 9IYK) display an asymmetric unit consisting of six dimers and twelve monomers. Each chain is colored differently, as indicated in the panel, and the coloring is consistent throughout this manuscript. (**B**) The B-factors of the hSOD1 structure are represented using ellipsoids colored in a rainbow spectrum, with blue color indicating the minimum value of 20.040 and red representing the maximum value of 204.990. The hSOD1 structure exhibits a gradient from the rigid core to the periphery (especially chains H and J), where the structure becomes more flexible. (**C**) The close-up view of the center of the structure reveals the symmetry around the Zn^CRYST^, with the catalytic domains containing Cu^2+^ and Zn^2+^ cations, as well as sulfate (SO_4_^2−^) and acetate (ACT) anions from the crystal conditions. (**D**) The Zn^CRYST^ brings four chains together, glued together through chain A and C’s His110 residues and chain B and D’s His-1 residues, which remain as an artefact after thrombin cleavage of the N-terminal hexahistidine tag. The *2Fo-Fc* electron density map is contoured at the 1σ level and colored in light blue. The distances between the coordinating His residues and Zn^CRYST^ are given in angstroms (Å).

**Figure 3 ijms-26-04228-f003:**
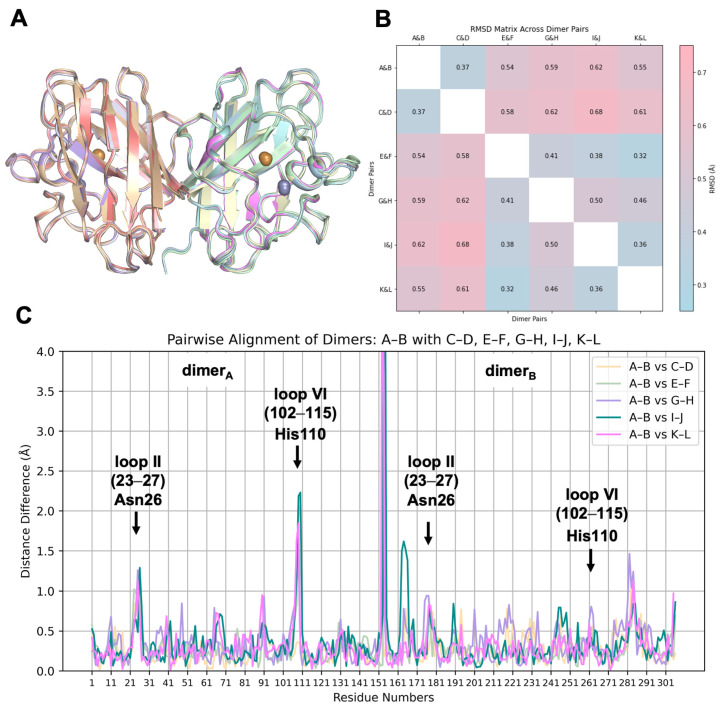
The superposition of six homodimeric hSOD1 molecules. (**A**,**B**) Superposition of the hSOD1 dimers shows moderate structural flexibility, with RMSD values ranging from a minimum of 0.32 Å to a maximum of 0.68 Å. (**C**) Pairwise alignment based on Cα atoms reveals notable structural shifts when dimer pair A-B is superimposed onto the remaining five dimeric pairs (C–D, E–F, G–H, I–J, and K–L). Specific regions, particularly loop II (residues 23–27) and loop VI (residues 102–115), exhibit RMSD values up to 2.5 Å, suggesting that these domains are intrinsically flexible. Comparison of the monomers, where the one on the left side is referred to as dimer_A_ (A, C, E, G, I, and K), while the one on the right is labeled dimer_B_ (B, D, F, H, J, and L). The most significant positional shift is observed at the N-terminal Ala1 of dimer_B_, with a displacement reaching 7 Å (shown up to 4 Å) due to residual and coordinated His-1 residues.

**Figure 4 ijms-26-04228-f004:**
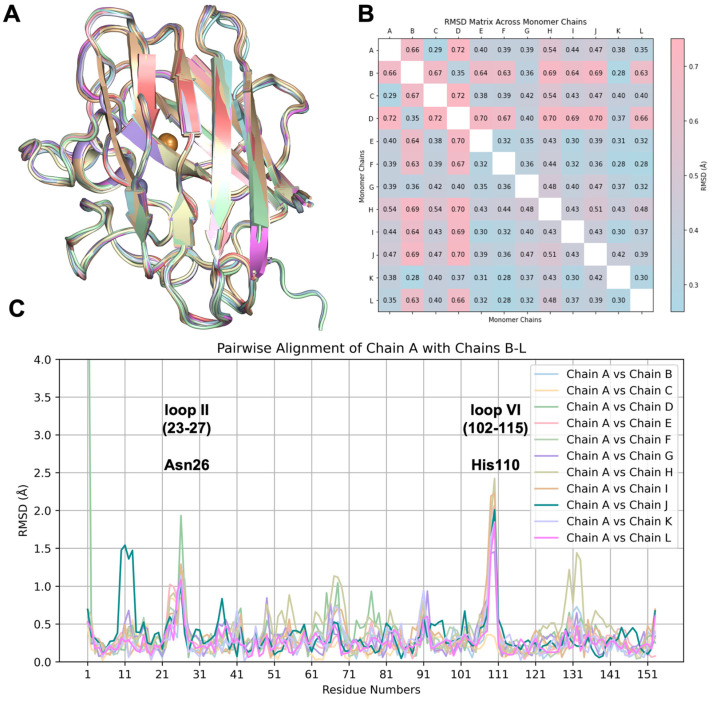
The superposition of twelve monomeric hSOD1 molecules. (**A**,**B**) Superposition of the hSOD1 monomers shows subtle structural plasticity, with RMSD values ranging from a minimum of 0.28 Å to a maximum of 0.72 Å. (**C**) Pairwise superposition of chain A against the remaining 11 chains (B, C, D, E, F, G, H, I, J, K, and L) reveals regions with modest variations. Notably, loop II (residues 23–27), particularly around Asn26, and loop VI (residues 102–115), especially around His110, show RMSD values up to 2.5 Å, indicating inherent flexibility. The first N-terminal residues of chains B and D exhibit the most pronounced shifts compared to other chains, with RMSD values reaching 7 Å (displayed up to 4 Å) due to their modified N-terminus containing extra residues.

**Figure 5 ijms-26-04228-f005:**
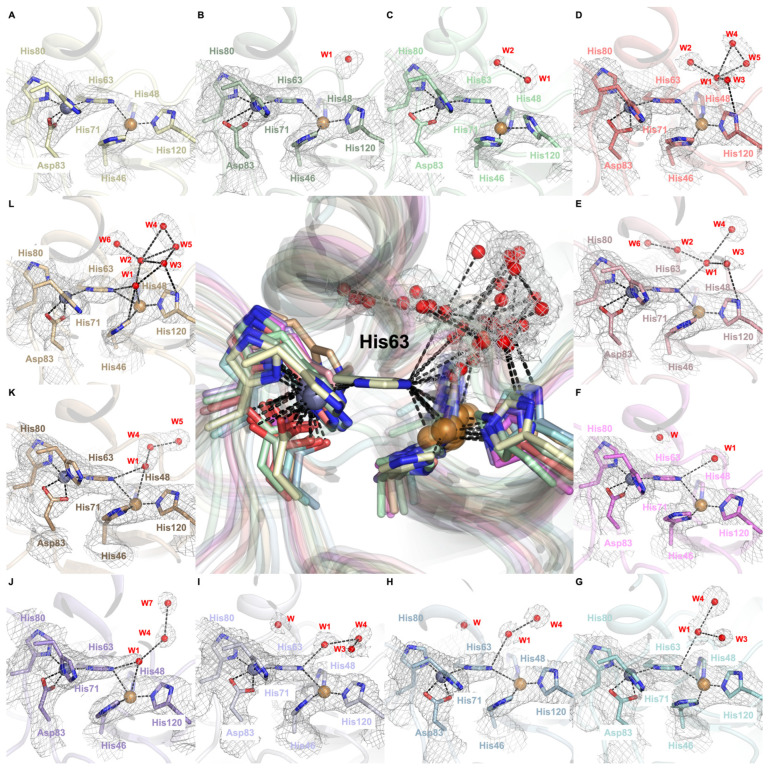
The water coordination of twelve hSOD1 active sites. The center panel displays the superpositions of all 12 active site regions, each centered on the CE1 atom of His63, revealing the positional shifts of water molecules (colored red) within the active site. Panels (**A**–**L**) illustrate the stepwise choreography of water molecules as they approach and depart from the catalytic site during the enzymatic cycle. The 2*Fo-Fc* electron density map, contoured at 1σ and colored in gray, highlights the positions of the water molecules and surrounding active site residues. The catalytic Cu^2^⁺ cation remains consistently coordinated with His63 through a strong covalent bond in all steps, except panel (**J**). W1 ultimately establishes coordination with Cu^2^⁺, forming the penta-coordinated native state in panels (**J**–**L**).

**Figure 6 ijms-26-04228-f006:**
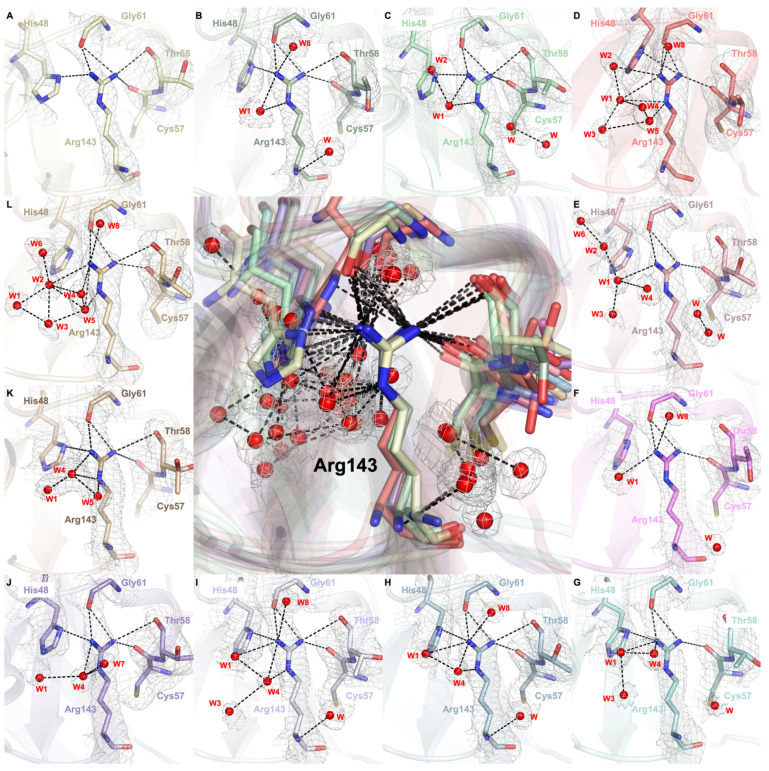
Allosteric water coordination around the Arg143 residue. The center panel displays the superpositions of all twelve Arg143 residues, each centered on the NE atom of Arg143, capturing the dynamic repositioning of water molecules interacting with Arg143 and its partner residues. Panels (**A**–**L**) depict the sequential movement of water molecules as they associate with and dissociate from Arg143 throughout the water steering mechanism. The 2*Fo-Fc* electron density map, contoured at 1σ and colored in gray, highlights Arg143, His48, Cys57, Gly61, Thr58, and the surrounding water network. Water molecules labeled with numbers are correspondent with the water molecules indicated in Figure 5, whilst water molecules labeled as “W” do not interact with the Arg143 side chain and do not have allosteric importance.

**Figure 7 ijms-26-04228-f007:**
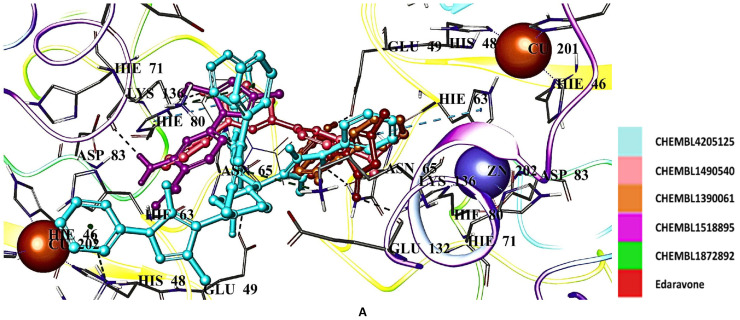
Docking poses (**A**) and interactions (**B**) of CHEMBL4205125, CHEMBL1490540, CHEMBL1390061, CHEMBL1518895, CHEMBL1872892, and edaravone in the binding pocket of hSOD1 (Blue dashes: π-π interactions and black dashes: hydrogen bonding). Histidine residues are labeled according to their three distinct protonation states, which depend on their net charge and the position of their proton(s). HIP carries a +1 charge with both δ- and ε-nitrogens protonated; HID is neutral, with only δ-nitrogen protonated; and HIE is also neutral but with protonation at the ε-nitrogen [58].

**Figure 8 ijms-26-04228-f008:**
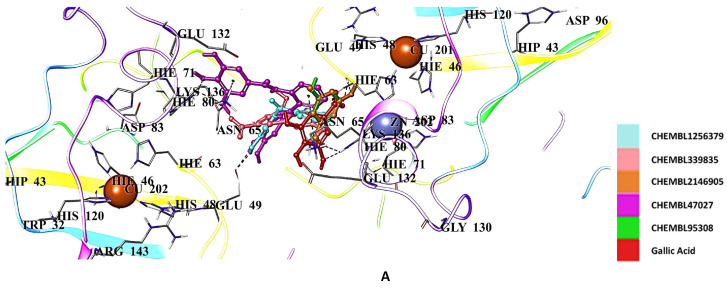
Docking poses (**A**) and interactions (**B**) of CHEMBL1256379, CHEMBL339835, CHEMBL2146905, CHEMBL47027, CHEMBL95308, and gallic acid at the binding site of hSOD1 (Blue dashes: π-π interactions and black dashes: hydrogen bonding).

**Figure 9 ijms-26-04228-f009:**
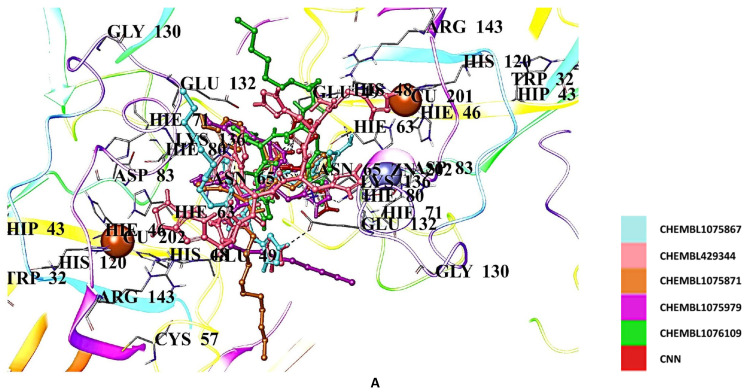
Docking poses (**A**) and interactions (**B**) of CHEMBL1075867, CHEMBL429344, CHEMBL1075871, CHEMBL1075979, CHEMBL1076109, and CNN at the binding site of hSOD1 (Blue dashes: π-π interactions and black dashes: hydrogen bonding).

**Figure 10 ijms-26-04228-f010:**
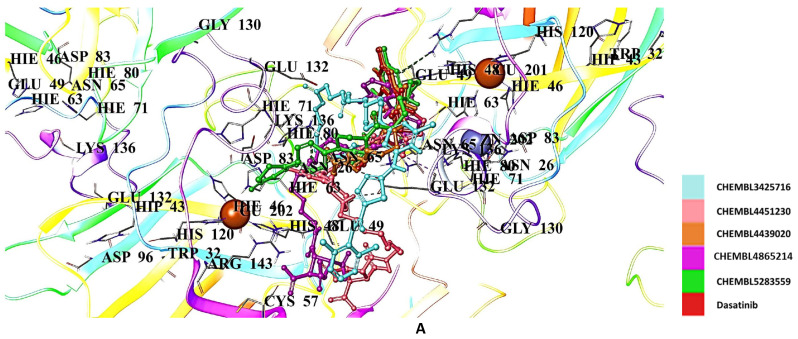
Docking poses (**A**) and interactions (**B**) of CHEMBL3425716, CHEMBL4451230, CHEMBL4439020, CHEMBL4865214, CHEMBL5283559, and dasatinib at the binding site of hSOD1 (Blue dashes: π-π interactions and black dashes: hydrogen bonding).

**Figure 11 ijms-26-04228-f011:**
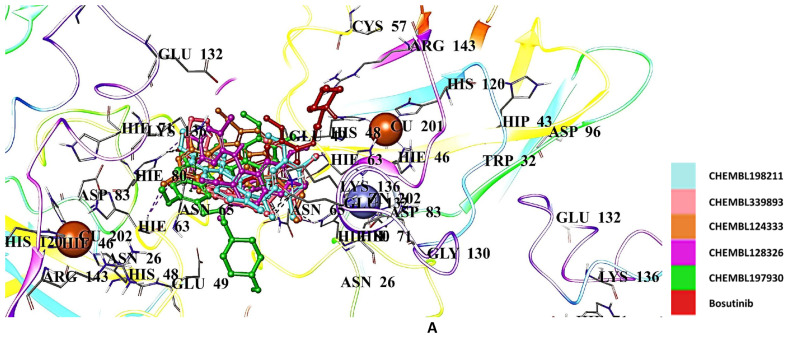
Docking poses (**A**) and interactions (**B**) of CHEMBL198211, CHEMBL339893, CHEMBL124333, CHEMBL128326, CHEMBL197930, and bosutinib at the binding site of hSOD1 (Blue dashes: π-π interactions and black dashes: hydrogen bonding).

**Figure 12 ijms-26-04228-f012:**
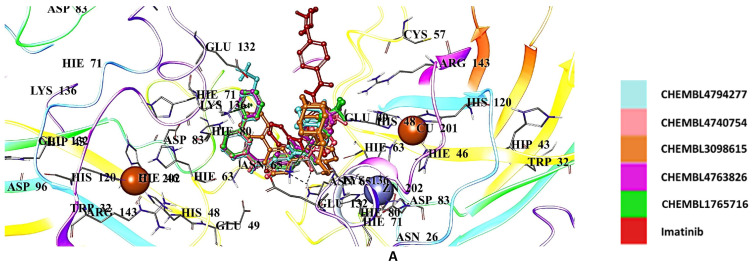
Docking poses (**A**) and interactions (**B**) of CHEMBL4794277, CHEMBL4740754, CHEMBL3098615, CHEMBL4763826, CHEMBL1765716, and imatinib at the binding site of hSOD1 (Blue dashes: π-π interactions and black dashes: hydrogen bonding).

**Table 1 ijms-26-04228-t001:** Data collection and refinement statistics.

PDB ID	9IYK
Data collection	
Beamline	Diamond Beamline i03
Space group	C 1 2 1
Cell dimensions	
a, b, c (Å)	178.19 138.25 112.93
α, β, γ (°)	90 129.2 90
Resolution (Å)2	56.47–2.34 (2.38–2.34)
CC(1/2)	0.94 (0.26)
I/σI	9.0 (0.5)
Completeness (%)	99.1 (97.6)
Redundancy	7.1 (7.3)
Refinement	
Resolution (Å)	48.92–2.34 (2.37–2.34)
No. reflections	77,004 (77,004)
Rwork/Rfree	0.20/0.26 (0.35/0.39)
No. non-hydrogen atoms	14,379
Protein	13,815
Ligand	73
Water	503
B-factors	
Protein	70.15
Ligand	64.69
Water	53.82
Coordinate errors	0.41
R.m.s deviations	
Bond lengths (Å)	0.004
Bond angles (°)	0.634
Ramachandran plot	
Favored (%)	96.64
Allowed (%)	3.30
Disallowed (%)	0.06

**Table 2 ijms-26-04228-t002:** Docking scores (kcal/mol) of compounds at the binding site of hSOD1.

Compound	Docking Score	Compound	Docking Score
CHEMBL4205125	−7.641	CHEMBL3425716	−7.244
CHEMBL1490540	−7.002	CHEMBL4451230	−7.231
CHEMBL1390061	−6.865	CHEMBL4439020	−7.164
CHEMBL1518895	−6.767	CHEMBL4865214	−6.972
CHEMBL1872892	−6.484	CHEMBL5283559	−6.889
CHEMBL1256379	−7.121	CHEMBL198211	−6.459
CHEMBL339835	−7.027	CHEMBL339893	−6.291
CHEMBL2146905	−6.794	CHEMBL124333	−6.166
CHEMBL47027	−6.778	CHEMBL128326	−6.156
CHEMBL95308	−6.761	CHEMBL197930	−6.142
CHEMBL1075867	−9.760	CHEMBL4794277	−6.527
CHEMBL429344	−8.935	CHEMBL4740754	−6.491
CHEMBL1075871	−8.858	CHEMBL3098615	−6.489
CHEMBL1075979	−8.841	CHEMBL4763826	−6.488
CHEMBL1076109	−8.757	CHEMBL1765716	−6.193
Edaravone	−4.831	Dasatinib	−5.751
Gallic acid	−5.428	Bosutinib	−4.382
CNN	−5.769	Imatinib	−4.969

## Data Availability

The atomic coordinates and structure factors have been deposited in the Protein Data Bank (PDB) under accession code 9IYK (Crystal structure of hSOD1 in the C121 space group).

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
