# Peer review of "Structural Insights into the Dynamics of Water in SOD1 Catalysis and Drug Interactions"

_ijms, 2025, doi:10.3390/ijms26094228_

Round 1

Reviewer 1 Report

Comments and Suggestions for Authors

This is an experimental-theoretical study. I´m going to discuss only the theoretical part, since that is my area of expertise. It was carried out with molecular docking, and deals with the interaction between SOD and some antioxidants. The role of water molecules inside the pocket was also considered. The manuscript is well written and presented in a clear and appealing way. An appropriate reviewing of the literature is provided, and the conclusions are well supported by the gathered evidence. Some future investigations are foreseen to extend the knowledge on this ingredient and to deepen the knowledge in this complex and interesting natural product.

Therefore, this is my opinion that the manuscript is in line with the scope of the IJMS and can be published in its present form.

However, I have some minor points for the authors to consider:

  • It would be very useful for the potential readers if the authors provide a table with the values of the union energies, and the IC50.
  • Refining of at least some of the structures using molecular dynamics would increase the reliability of the provided data.

Author Response

Dear Editor,

We would like to thank Reviewer 1 for their invaluable comments on the structure of our manuscript. You can find our response to your comments below.

Reviewer 1

This is an experimental-theoretical study. I´m going to discuss only the theoretical part, since that is my area of expertise. It was carried out with molecular docking, and deals with the interaction between SOD and some antioxidants. The role of water molecules inside the pocket was also considered. The manuscript is well written and presented in a clear and appealing way. An appropriate reviewing of the literature is provided, and the conclusions are well supported by the gathered evidence. Some future investigations are foreseen to extend the knowledge on this ingredient and to deepen the knowledge in this complex and interesting natural product.

Therefore, this is my opinion that the manuscript is in line with the scope of the IJMS and can be published in its present form.

However, I have some minor points for the authors to consider:

It would be very useful for the potential readers if the authors provide a table with the values of the union energies, and the IC50.

  • As the Reviewer requested, the docking scores of hit compounds were added as Table 2.
  • The experiments were performed mainly against mutant SOD1 mice and most of the experiments included neuron survival, weight loss, Rotarod, and the grip power test results. Therefore, the results were not indicated with IC50 values. However, as the Reviewer requested, we added more detailed quantitative data for each standard agent indicated below:

Edaravone: Ito et al. 2008 [33] determined the in vivo efficacy of edaravone for possible future treatment of ALS with the first randomized blind study. Their findings indicated that 15 mg/kg edaravone administration lowered the rate of reduction of parameters such as the body weight, the mean persistence time on the Rotarod, and the grip power in SOD1G93A mutant mice. One of the most striking points they reached was the mean area of abnormal SOD1 deposition per anterior horn and its root exit zone were decreased significantly. In other study, Aoki et al. 2011 [34] reported that edaravone improved motor functions in SOD1H46R mutant male rats with  5.6 ± 0.5 cm mean landing foot-splay distance between the hind feet at high-dose group.

Gallic acid: They also showed that gallic acids were able to bind to the soluble SOD protein (12–44 μM kd) indicating its direct binding to the SOD1 protein [41].

CNN: Noguchi et al. 2019 [46] reported that L-carnosine hydrazide (L-carnosine-NHNH2, CNN) (Figure 1) exhibited protective effects in PC-12 cells with 30 mM concentration towards 250 μM 4-HNE and abolished delayed neuronal death by >60% in the pre-treated group and by >40% in the post-treated group.

Dasatinib: Dasatinib (Figure 1) showed neuroprotection in SOD1G93A transgenic ALS mice reducing the cytotoxicity of mutant SOD1 significantly based on cell viability and cell death assays (P<0.05). Dasatinib also improved survival, the weight loss, and a poor grip strength of SOD1G93A mice at a dose of 25 mg/(kg·day) compared with vehicle treatment (P<0.01, 25 mg/(kg·day)) [50].

Imatinib: Imatinib (Figure 1) showed micromolar inhibition of Abl1 phosphorylation in primary neuron cultures in response to oxidative stress. Rojas et al. 2015 [56] revealed that 2 μM imatinib along with astrocyte conditioned media (ACM)-SOD1G93A alleviated the intensity of immunoreactivity for phosphorylated c-Abl and the number of positively stained cells [56].

Bosutinib: Imamura et al. 2017 [57] performed high-throughput screening of numerous compounds using survival of ALS patient iPSC-derived motor neurons as readout. They determined that more than half of the hits targeted the Src/c-Abl signaling pathway and they identified that bosutinib (Figure 1) increased ALS motor neuron survival (mean survival: 164.1 ± 9.4 days compared to vehicle (156.3 ± 8.5 days)) and regulated misfolded SOD1 proteins in transgenic mice.

Refining of at least some of the structures using molecular dynamics would increase the reliability of the provided data.

  • Thanks for the kind suggestion from the Reviewer. We would like to update the latest version of Schrodinger, but now we have a limited budget for this process. The old version does not allow us to perform MD through the Schrodinger-Desmond module and we have also reached our free limits for academic users. Another problem is that we are not able to use other MD programs since we have been using Desmond for a long time. Our efforts in the development of anti-ALS drug candidates on the basis of computer-aided drug design are continuing, and we would like to add MD experiments combined with co-crystallization with drug candidates to our future studies. We hope the Reviewer understands our circumstances.

The revised manuscript has been submitted to International Journal of Molecular Sciences. We look forward to hearing your positive response.

Reviewer 2 Report

Comments and Suggestions for Authors

The manuscript presents a new structural study of human Cu-Zn superoxide dismutase 1 (hSOD1). The study revealed the dynamic nature of SOD1 nature, and explored the role of water molecules in the active site, their influence on proton transfer, and how structural dynamics contribute to SOD1 function. In addition, it also investigated molecular docking of radical scavengers and Abl1 inhibitors targeting misfolded SOD1, highlighting a new chemical, CHEMBL1075867, to be a promising blueprint for new drug development for ALS.

Overall, I think this study provides valuable structural insights into SOD1.The findings are well-supported by experimental data. I support the publication for this manuscript at IJMS. I only have one minor concern.

The new crystal form is likely an outcome only relevant to this study, as the deletion of His-1 or extra Zn2+ will disintegrate this structure. To eliminate the confusion, the authors should add a few sentences in the discussion section, to discuss the scope of the relevance of the structure.

Author Response

Dear Editor,

We would like to thank Reviewer 2 for their invaluable comments on the structure of our manuscript. You can find our response to your comments below.

Reviewer 2

The manuscript presents a new structural study of human Cu-Zn superoxide dismutase 1 (hSOD1). The study revealed the dynamic nature of SOD1 nature, and explored the role of water molecules in the active site, their influence on proton transfer, and how structural dynamics contribute to SOD1 function. In addition, it also investigated molecular docking of radical scavengers and Abl1 inhibitors targeting misfolded SOD1, highlighting a new chemical, CHEMBL1075867, to be a promising blueprint for new drug development for ALS.

Overall, I think this study provides valuable structural insights into SOD1. The findings are well-supported by experimental data. I support the publication for this manuscript at IJMS. I only have one minor concern.

The new crystal form is likely an outcome only relevant to this study, as the deletion of His-1 or extra Zn2+ will disintegrate this structure. To eliminate the confusion, the authors should add a few sentences in the discussion section, to discuss the scope of the relevance of the structure.

  • We would like to thank the reviewer for their valuable suggestions and for raising this important point. We have added the following sentences in the discussion section to clarify the relevance and scope of the structure: 

This zinc-mediated dimer-dimer interaction arises in crystallo, predominantly driven by exogenous Zn2+ and the artefactual His-1 residue originating from the recombinant N-terminal hexahistidine tag. As a result, removing either His-1 or excess Zn2+ would likely disrupt this specific crystallographic arrangement. Nevertheless, this observation emphasizes the intrinsic capacity of SOD1 to engage in non-native metal coordination, highlighting structural flexibility and potential transient interactions, relevant under physiological or pathological conditions. Importantly, residues participating in this zinc coordination, particularly the surface-exposed His110, are distant from the active site. Thus, the coordinated water dynamics within the catalytic region remain undisturbed, preserving the integrity of the observed water-mediated conformational changes. 

The revised manuscript has been submitted to International Journal of Molecular Sciences. We look forward to hearing your positive response.